# Quality of Life of Emirati Women with Breast Cancer

**DOI:** 10.3390/ijerph20010570

**Published:** 2022-12-29

**Authors:** Linda Smail, Ghufran Jassim, Sarah Khan, Syed Tirmazy, Mouza Al Ameri

**Affiliations:** 1College of Interdisciplinary Studies, Zayed University, Dubai 19282, United Arab Emirates; 2Department of Family Medicine, Royal College of Surgeons in Ireland-Medical University of Bahrain, Busaiteen 15503, Bahrain; 3College of Natural and Health Sciences, Zayed University, Dubai 19282, United Arab Emirates; 4Oncology Center, Dubai Hospital, Dubai 7272, United Arab Emirates; 5Breast Cancer Center, Tawam Hospital, Al Ain 15258, United Arab Emirates

**Keywords:** breast cancer, emirati women, quality of life, EORTC QLQ-BR23, EORTC QLQ-30

## Abstract

To examine the quality of life (QoL) of Emirati women with breast cancer (BC) and determine its relationships with their sociodemographic characteristics and clinical factors. The study will play a leading role in providing information about the QoL of Emirati women with BC and will help in recognizing the aspects of QoL in BC survivorship that requires special attention. A population-based cross-sectional study was conducted on 250 Emirati women using a multistage stratified clustered random sampling. The participants were interviewed face-to-face using a structured questionnaire composed of sociodemographic variables, reproductive characteristics, and the European Organization for Research and Treatment of Cancer Quality of Life Cancer-Specific version (EORTC QLQ-C30, v.3.0) and the EORTC QoL Breast Cancer-Specific version (EORTC QLQ-BR23) translated into Arabic. Emirati BC survivors reported good QoL overall. The most bothersome symptoms were sleep disturbance, fatigue, pain, hair loss and arm symptoms. Emirati women scored average on all functional scales, which indicates mediocre functioning, but high on the symptom scales, which indicates worse symptoms. Factors associated with a decline in the domains of QoL included higher age, lower income, and history of metastases, mastectomy, and lymph node dissection.

## 1. Introduction

Female breast cancer (BC) is the leading cause of death among women and one of the leading cancer types worldwide in terms of the number of new cases since 2018 [1]. Many efforts have been made over the last decade not only to understand, prevent, diagnose, and treat BC but also to improve the quality of life (QoL) of BC survivors.

Surviving BC can have a very large impact on survivors’ lives. People often presume that surviving the disease is the end of the battle. What they do not know is that survivors face severe challenges, both emotional and physical. Some of the symptoms usually include depression, pain, fatigue, loss of interest in sexual activities, low self-esteem and many more symptoms [1].

BC is the most common cancer among Arab women and represents between 14% and 42% of all female cancers. Moreover, approximately 50% of the reported cases in the United Arab Emirates (UAE) are women younger than 50 years, compared to only 25% in other developed countries [2]. According to the data from the American Institute of Cancer Research, the number of new BC cases in 2018 represented 39.9% of all cancer types among females in the UAE (Emirati and non-Emirati), with an associated mortality rate of 12.4%. The cumulative risk of BC in the UAE has increased for years, reaching 5.8% in 2018 [3].

There is a large amount of research on the QoL of women with BC in Western societies, and this research has identified several factors affecting the QoL of women with BC [4,5,6,7,8,9,10,11,12,13,14]. The results have shown that BC survivors report a moderate to high prevalence of psychological morbidities such as depression, anxiety, pain and sleeping disorders. However, research in the Arab world is very limited. Recent reviews [15,16] studied the QoL of women with BC in the Middle East (ME) and found that less than one-third of the study participants from the ME had a good QoL.

The few existing studies in the UAE [4,17,18,19,20] showed that women in the UAE (Emirati and non-Emirati) have little to poor knowledge about BC. This low level of knowledge is coupled with social, cultural and religious restrictions [18], resulting in low uptake of BC screening services. However, to our knowledge, very few studies have assessed QoL among Emirati women with BC, this could be due to many factors such as: Arab women in general face cultural taboos surrounding BC as some families fear that their daughters will not get married if the mother’s diagnosis of BC becomes known [18,19], the sensitivity of this part of women’s body, the conservative and religious nature of the community, and the restrictions in granting ethics approval for breast cancer studies. With the development in education and the participation of Emirati women in the workforce, there is more engagement from the Emiratis in women’s health, and therefore more ethics clearance for studies addressing such a sensitive topic.

The study will help in recognizing the aspects of QoL in BC survivorship that lack assets and provide evidence-based data that can help solve problems related to surviving BC for Emirati women.

## 2. Materials and Methods

### 2.1. Study Design and Sample

A community-based cross-sectional study was conducted from September 2020 to April 2021 among 250 Emirati women. The study participants were recruited from two hospitals in the UAE: Dubai Hospital in the Emirate of Dubai and Tawam Hospital in Al Ain city. The sample size was determined based on a power of 85% and a significant level of 5% with an estimated non-response rate of 20%.

### 2.2. Sampling Method

Dubai Hospital is the only governmental hospital in Dubai with an oncology center, while Tawam Hospital is considered the cancer referral governmental hospital in the UAE. While Dubai Hospital falls under the Dubai Health Authority, Tawam Hospital falls under the Department of Health of the Emirate of Abu Dhabi, the capital of the UAE.

The lists of patients in both Dubai and Al Ain hospitals were obtained; all registered women were contacted by phone to either administer the survey or determine their interest to take the survey face to face during their next visit. Non Registered women visiting for consultation or visiting the cancer care unit were also approached to take part in the study through face-to-face interviews.

The list obtained from the Dubai Hospital Cancer Registry had 212 registered women since 2016. Out of these Emirati women, 69 had survived BC, while 13 had not. The list from Tawam Hospital had 113 registered women, 75 registered from 2020 and 38 registered as of August 2021. The response rate from Dubai Hospital was 100%, as all 69 Emirati women registered in the hospital took the survey, while the response rate from Tawam Hospital was 66.4%. As the subject was very sensitive, nurses trained by a specialist in BC interviewed the women by explaining the study and asking them all the questions in the questionnaire. The study was carried out from August 2020 to April 2021. Emirati women with BC were included in the study while Emirati women with another type of cancer were excluded.

### 2.3. Study Instruments

A structured questionnaire was administered through face-to-face interviews in the hospitals. The questionnaire included three parts. The first part collected information on sociodemographic and reproductive characteristics, including age, level of education, marital status, employment, and smoking habits. It also collected BC data, such as time elapsed since diagnosis, according to which women were defined as having an early diagnosis (≤1 year since diagnosis), being in the transitional period (>1 and ≤5 years since diagnosis) and being long-term survivors (>5 to ≤10 years).

The second part assessed the QoL of Emirati women with BC using the European Organization for Research and Treatment of Cancer Quality of Life Cancer-Specific version (EORTC QLQ-C30, v.3.0).

The third part contained the Quality of Life Breast Cancer-Specific version (EORTC QLQ-BR23), which explores five domains: therapy side effects, arm symptoms, breast symptoms, body image, and sexual functioning.

The EORTC QLQ-C30 is a 30-item instrument designed to assess cancer patients’ physical, psychological and social functioning [12,21]. It is composed of nine multi-item scales: five functional scales, a global QoL scale, and three symptom scales (fatigue, pain, and nausea and vomiting). In addition, there are five single-item symptom scales (dyspnea, sleep disturbance, appetite loss, constipation, and diarrhea) and a final item that evaluates the perceived financial impact of the disease.

Each of the first 28 items of this instrument is answered on a scale from 1 (not at all) to 4 (very much). The time frame is the present moment. For item number 29 (on overall general health) and item 30 (on overall QoL), the response options range from 1 (very poor) to 7 (excellent), and the time frame is during the past week.

The QLQ-BR-23 [12,22] consists of 23 items divided into two multi-item functional scales (body image and sexual functioning), three symptom scales (systemic side effects, breast symptoms, and arm symptoms), and three single-item scales on sexual enjoyment, future perspectives, and upset by hair loss. The response options are on a scale of 1 (not at all) to 4 (very much), and the time frame is during the past week, except for the sexual items (during the past four weeks).

### 2.4. Validity and Reliability of the Questionnaire

The Arabic versions of the QLQ-C30 and QLQ-BR23 were developed and translated by the EORTC. The validation of the QLQ-C30 and QLQ-BR23 in the Arab women living in the UAE was performed by Awad et al. [4].

A professional translator translated the English version of the sociodemographic part into the Arabic language. A second bilingual speaker checked the Arabic version word by word with the English version and then translated it back to English (World Health Organization Translation process). The content validity of the Arabic version of the questionnaire was assessed by a panel of experts in the field to evaluate the readability, language simplicity and suitability of the items and to evaluate the relationship of each item to the whole scale. The panel was composed of a professor in obstetrics and gynecology, an oncologist, and a public health specialist. Based on their comments, changes were made. The internal consistency reliability of the Arabic version of the questionnaire of the current study was assessed using Cronbach’s α, with coefficients of 0.918 and 0.878 for the EORTC QLQ-C30 and QLQ BR-23, respectively, which suggested relatively good internal consistency.

### 2.5. Ethical Consideration

Ethical clearance was obtained from the Zayed University Research and Ethics Committee [ZU20_013_F], Dubai Health Authority Medical Research Committee [DSREC-01/2020_22], and Abu Dhabi Health Research and Technology Ethics Committee [DOH/CVDC/2021/327]. Institutional review boards at participating institutions approved procedures and protocols. All procedures performed in this study were in accordance with the ethical standards of the institutional and national research committees and with the 1964 Helsinki declaration and its later amendments or comparable ethical standards. All data sampling and collection methods were performed in accordance with the relevant guidelines and regulations provided by the above mentioned institutions. 

The Emirati women on the hospital lists were invited to participate in the study on a voluntary basis. The study purpose was explained to the patients, and informed consent was obtained from participants and from legal guardians of illiterate participants. Consent was obtained for all forms of personally identifiable data including biomedical, clinical, and biometric data. However, confidentiality and privacy were maintained, and any identifiable data was numerically coded so that it could not be traced back to the participant. No organs or tissues were obtained for this study.

### 2.6. Statistical Analysis

The collected data were coded, entered, and analyzed using the statistical package SPSS version 26 (IBM Corp., Armonk, NY, USA). Descriptive statistics were computed to describe all items of the questionnaire. Scores were calculated for both the EORTC QLQ-C30 and EORTC QLQ-BR23 patient responses using a scoring algorithm recommended by the EORTC [22,23]. The scoring algorithm involves first computing the average of the item responses and second transforming the score to a value on a 0–100% scale.

A high scale score represents a higher response level. Hence, a high score for a functional scale represents a high level of functioning, and a high score for the global health status/QoL represents a high QoL; however, a high score for a symptom scale item corresponds to more frequent and/or more intense symptoms.

For the functional scales and the global QOL, subjects with problematic functioning were defined as those who scored below 33.3% (<33.3%), while subjects in good condition were those who scored 66.7% or more (≥66.7%). For symptom scales, subjects who scored below 33.3% were classified as having less severe symptoms, while those scoring 66.7% or more were classified as having more intense symptoms [24].

One-way analysis of variance (ANOVA) or the independent-sample t test were carried out to test the equality of population means across the categories of each independent variable (predictor) depending on the number of categories for the independent variables. In case the statistical assumptions required for one-way ANOVA and t test were violated, nonparametric tests, namely, Kruskal–Wallis and Mann–Whitney, were used instead. Additional exploration of the differences among the means was determined by post hoc analysis.

Pearson’s linear correlation coefficient was computed to assess the linear relationship between each of the outcome variables and each of the quantitative independent variables. Global health status/QoL and the functional and symptom scales served as the dependent variables. The independent variables (age, duration since diagnosis, marital status, educational level, employment status, income, menopausal status, pathological staging, history of metastases, chemotherapy, lumpectomy, mastectomy, lymph node dissection, radiotherapy, and hormonal therapy) were categorized into two categories (yes and no) and served as predictors for the multiple linear regression models. Adjusted R^2^ was computed, and statistical tests with *p* < 0.05 were considered statistically significant.

To ensure there were no multicollinearity, Pearson correlation coefficients were calculated to examine the relationships between the predictors. The coefficients (min r = 0.02 and max r = 0.47) suggested that the assumptions of multicollinearity were not violated. Moreover, tolerance variance inflation factor (VIF) values did not indicate a violation of this assumption.

A Durbin-Watson statistic was calculated to assess the assumption that the values of residuals are independent, which suggested that this assumption was not violated in all models.

A Scatter plot was created to assess the assumption that the variance of the residuals was constant (homoscedasticity). Furthermore a P-P plot was created to assess the assumption that the values of the residuals are normally distributed. Additionally, Cook’s distance values were calculated to ensure that no influential cases were biasing the models. 

## 3. Results

### 3.1. Characteristics of the Study Sample

The mean age of the 250 Emirati women was 53.4 (SD ± 11.3), and the range was 59 (27 was the minimum and 86 the maximum). The mean time elapsed since diagnosis was 4.44 (SD ± 4.3), with a minimum of 0 (year 2021) and a maximum of 25 years (year 1996) (Table 1). It is important to note that all these reported values are not the absolute sum of scores, but the relative sum of scores in a scale of 0–100% as recommended by the EORTC [22,23].

Out of the 250 participants, 99 (39.6%) were diagnosed in 2020, while 45 (18%) were diagnosed more than 5 years ago.

The majority of women (70%) were from Dubai Hospital, while 30% were from Tawam Hospital. More than half of the participants (56.8%) were from Dubai, 19.6% were from Abu-Dhabi, 12% were from Sharjah, and 11.6% were from the other emirates (Ajman, Ras al Khaimah, Um Al Quwain, and Fujairah).

The majority of women (95.2%) never smoked, and 87.6% of them had children, with a mean number of 4 (SD ± 3) children (a maximum number of 10 children). Approximately 39% of the participants exercised on a daily basis, and approximately 4% lived alone. In total, 44% of them had had a miscarriage, with a mean number of 1 (SD ± 1.3) miscarriages (a maximum number of 6 miscarriages). More than a quarter (25.6%) of the participants had a family history of BC, and 11.6% had the cancer spread to other parts of their body.

### 3.2. Quality-of-Life Scale Scores

The global health/QoL mean score of the 250 participants was 74.73 (SD ± 18.25) indicating a good level of wellbeing (Table 2).

On the QLQ-C30 scales, the mean scores for the five functional scales ranged from 68.43 to 82.33, showing mostly a good level of functional health status. While social functioning scored the highest (82.33 ± 28.38) among the functional scales, emotional functioning scored the lowest (68.43 ± 30.02).

On the symptom scales, the most worrying symptom was sleep disturbance (47.87 ± 38.46), followed by fatigue (38.18 ± 30.31) and pain (29.13 ± 28.01). Financial impact scored the lowest, indicating that most women did not have financial issues related to their cancer. Out of the 250 participants, 5.6% to 12.4% had problematic functioning on the functional scales but worse functioning on the symptom scales, as 6.8% to 45.6% had problems with symptoms.

On the QLQ-BR23 scales, the range of the mean scores for the functional scales was 50.80 to 80.30, showing mostly above average to good levels of functional health status. While sexual functioning scored the highest (86.07 ± 22.61) among the functional scales, future perspective scored the lowest (50.80 ± 37.92).

On the symptom scales, the most worrying symptom was upset by hair loss (61.01 ± 37.35), followed by arm symptoms (33.73 ± 28.08). Notably, all symptom scales means had 0 as the minimum score and 100 as the maximum score, with the exception of systemic side effects, for which the maximum was 95.42. Out of the 250 participants, 2% to 24% had problematic functioning on the functional scales but worse functioning on the symptom scales, as 12% to 40% had problems.

### 3.3. Factors Associated with Quality-of-Life Scale Scores

#### 3.3.1. Global Health, Functional, and Emotional Scales on the QLQ-C30

Table 3 shows that there were significant differences in the global health/QoL means across categories of monthly income (*p* = 0.018), physical activity (*p* = 0.0004), history of metastases (*p* = 0.001), and type of treatment (*p* = 0.045 for chemotherapy). Post hoc analysis showed that participants who had regular physical activity, had high income, had no history of metastases, and were not treated with chemotherapy seemed to have better global health-related QoL.

Furthermore, significant differences in the physical functioning means were observed across categories of monthly income (*p* = 0.007), physical activity (*p* < 0.001), history of metastases (*p* = 0.027), and disease stage (*p* = 0.025). Post hoc analysis showed that participants who had regular physical activity, had high income, had no history of metastases, and were in early pathological staging had better functioning on the physical functioning scale.

Differences in the emotional functioning means were observed across categories of age, educational level, menopausal status, lymph node dissection (*p* < 0.0001 each), time since diagnosis (*p* = 0.002), marital status (*p* = 0.001), employment status (*p* = 0.014), and monthly income (*p* = 0.018). Post hoc analysis showed that participants who were aged above 50, were long-term survivors, were not employed, went through normal menopause, and had no history of lymph node dissection had better emotional functioning (Table 3).

#### 3.3.2. Symptom Scales on the QLQ-C30

With the exception of appetite loss and constipation, there were significant differences in all symptom scales across age categories. Women aged below 50 had worse symptoms on the symptom scales than those aged above 50. Moreover, there were significant differences in the mean pain scores by time since diagnosis (*p* = 0.026), pathological staging (*p* = 0.007), menopausal status (*p* = 0.010), education level (*p* = 0.012), monthly income (*p* = 0.005) and metastasis categories (*p* = 0.035). Post hoc analysis revealed that those who were older, were long-term survivors, were postmenopausal, and had a minimum education level (primary) experienced more pain.

Furthermore, there were significant differences in financial impact across age, menopausal status, monthly income, lymph node dissection, and chemotherapy. Post hoc analysis revealed that those who were older, postmenopausal, had an average salary, and had undergone lymph node dissection or chemotherapy experienced more financial impact.

#### 3.3.3. Functional and Symptom Scales on the QLQ-BR 23

As shown in Table 4, differences in the means of body images were significant among the categories of all independent variables with the exception of a history of metastases, radiology, and lumpectomy. Post hoc analysis showed that those who were younger, were employed, were premenopausal, were single, had low income, had undergone chemotherapy or dissection and were highly educated seemed to have poorer body image.

Better sexual functioning was observed for women who were aged above 50 (*p* < 0.001), were long-term survivors (*p* = 0.001), were not working (*p* < 0.001) and had gone through natural menopause (*p* < 0.001). Furthermore, post hoc analysis showed that women aged above 50 and those who had radiology as treatment tend to have better sexual enjoyment functioning.

More intense upset by hair loss was noted among women who were aged above 50 (*p* = 0.047).

Additionally, women aged above 50 had worse systemic side effects (*p* < 0.0001) and breast (*p* < 0.001) and arm (*p* = 0.001) symptoms.

Women who had surgical menopause complained of more severe systemic side effects (*p*= 0.017) and breast (*p* = 0.006) and arm (*p* = 0.019) symptoms than women who had menopause naturally.

Women who were recently diagnosed complained of more severe arm symptoms (*p* = 0.001), and women in advanced pathological staging complained of more severe systemic side effects.

#### 3.3.4. Predictors of Quality of Life

Table 5 below summarizes the adjusted regression models for the QLQ-C30.

As shown in Table 5, the predictors explained 6.3% of the variation in global health, 16.9% of the variation in emotional functioning, 19.2% of the variation in cognitive functioning, and 14.7% of the variation in social functioning. Monthly income was the only predictor that had a significant effect on global health/QoL given the other predictors in the model (*p* = 0.002). While age was significant only in emotional functioning (*p* = 0.004), education was a significant predictor in the emotional and social functioning models (=0.008 and 0.019, respectively). Metastases and mastectomy were significant only in the cognitive functioning model (*p* = 0.008 and 0.019, respectively). Lymph node dissection was significant in the emotional, cognitive, and social functioning models (*p* = 0.029, 0.017, and 0.027, respectively).

The important results of the regression analysis of the QLQ−C30 symptoms scales are that predictors explained 10.1% of fatigue, 8.7% of pain, and 16.5% of insomnia. Additionally, radiology was the only significant predictor in explaining fatigue, education was more important in explaining pain (*p* = 0.035), and advanced staging was important in explaining insomnia (Table of all results can be found in the Appendix A).

From the linear regression model with parameter estimates for the QLQ−BR23 functional and symptom scales (Table A1 and Table A2 Appendix A), the predictors explained between 8.7% and 31.8% of variation in all scales of the QLQ−BR23 with the exception of the upset by hair loss symptom. Late survivor was the only predictor that had a significant effect on breast symptoms given the other predictors in the model (*p* = 0.009), while radiology was the only predictor that had a significant effect on arm symptoms (*p* = 0.006).

Age and employment were significant predictors of sexual functioning (*p* < 0.0001 and 0.018, respectively). High income and mastectomy were significant predictors in the sexual enjoyment model (*p* = 0.020 and 0.044, respectively).

## 4. Discussion

This study is one of the few studies exploring the QoL of Emirati women with BC, and it revealed that Emirati women survivors of BC have good QoL and functioning but worse symptom experience. The most worrying symptoms for Emirati women were sleep disturbance, fatigue, and pain. A history of metastases and chemotherapy had a major effect across the domains of QoL in Emirati women.

Similarly, on the specific disease tool QLQ−BR23, the Emirati women seemed to perform above average to very good on the functional domains and poorly on the symptom scales. The most worrying symptoms were upset by hair loss and arm symptoms.

### 4.1. Comparison with Previous Literature

The mean global health/QoL of Emirati women with BC was higher (74.73) than that of other women in the Gulf Corporate Council (GCC). For example, the mean global health score was 63.9 in Bahrain [24], 64 and 67.45 in the KSA [25,26], and 45.3 in Kuwait [27]. Globally, this score is comparable to western women especially when compared to those in Europe and North America [28]. QoL scores reported from the West showed constant and steady increase in the last decade reaching up to 75 on a scale of 100 [29]. This could be attributed to many factors such as early diagnosis, improved treatment modalities, fewer comorbidities, older age at diagnosis and type of operation (breast preservation vs. mastectomy) when compared to the Arab world.

This variability could be attributed to different study populations, different tools used to measure the outcome, different sampling techniques and different times elapsed since diagnosis as well as poorly developed literature in this part of the world. For example, in a systematic review of 22 Arab countries, 13 studies only were identified addressing QoL of women with breast cancer [30].

Emirati women seemed to perform well on the five functional scales, with means ranging from 68.43 to 82.33, showing mostly a good level of functional health status. Social functioning scored the highest among Emirati women, which was similar to Bahraini, Saudi, and Kuwaiti women. Emotional functioning scored the lowest among Emirati women, which was similar to Bahraini women [24], but different from Kuwaiti and Saudi women, who scored lowest in physical functioning [26,27]. Emirati women had good functioning, with only 5.6% to 12.4% having problematic functioning on the functional scales, which was similar to Kuwaiti women (5.8% to 11.8%) [27] but better than Bahraini women (3.8% to 21.8%) [24].

Emirati women did worse on the symptom scales, as they scored higher on all symptom scales except financial impact. Indeed, while 1.7% to 17% of Bahraini women [24] were found to have had bad symptom experience, this figure was 8.8% to 45.6% for Emirati women. This may be explained by the fact that the time interval, for Emirati women, between initial experience of BC symptoms, and seeking medical help was between three months to three years [18].

Regarding financial impact, Emirati women were much better off than other Arab women in the region, with a score of 9.2, which is much lower than the scores of Bahraini women (34.58), Kuwaiti women (31.2), and Saudi women (17.13). This could be attributed to the fact that healthcare in the UAE is known to be among some of the finest in the world, offering a high standard of medical care in state−of−the art facilities. Healthcare facilities are run by the Dubai Health Authority (DHA), which oversees both public and private healthcare and replaces the Department of Health and Medical Services [31]. Health insurance is mandatory for all nationals and UAE residents. For UAE nationals, public hospitals and clinics are low−cost to free.

The most worrying symptom for Emirati women was sleep disturbance, followed by fatigue and pain, which was similar to studies on women in Bahrain, Kuwait and Saudi Arabia [24,26,27].

In accordance with the results from the QoL general scale, on the disease−specific scales (QLQ−BR23), Emirati women were above average to very good on the functional scales and poor on the symptom scales, as 12.4% to 40% had bad experience with BC symptoms, which was slightly similar to Kuwaiti women (6.7% to 40.8%), [27] but worse than Bahraini women (1.7% to 14.2%) [23] and Saudi women (9.5% to 26.8%) [26].

Among functional scales, sexual functioning scored the highest among Emirati women, which indicates better functioning. This was similar to Kuwaiti women but contrary to Bahraini and Saudi women, who scored the lowest. This finding should be interpreted with caution, as sexual functioning and enjoyment were perceived and approached differently in various studies due to the sensitivity of the topic and the conservative nature of the community. This scale showed the lowest reliability in a study investigating the reliability and validity of the Arabic versions of the EORTC QLQ−C30 and QLQ−BR23 questionnaires and had to be removed from the reliability analysis because of the very low coefficient values [32]. Compared to western women, Emirati women like many other Arab, Muslim women, treat their sexual life very privately and are hesitant to discuss it even with health care professionals and often described as shameful, impolite or forbidden [33].

For Emirati women with BC, the most worrying symptom was upset by hair loss, followed by arm symptoms, which was similar to all other Arab women in the region [24,25,26,27,30,34]. These topics should be given special care and attention. Physiotherapy and hair care options should be discussed and included in the comprehensive rehabilitation care provided to patients.

### 4.2. Factors Associated with Quality-of-Life Scores

The results of the study indicated associations between global health/QoL and monthly income, physical activity, history of metastases, and chemotherapy. Emirati women who had regular physical activity, had high income, had no history of metastases, and were not treated with chemotherapy seemed to have better global health−related QoL. Age was not associated with global health/QoL, which was similar to what was found in Bahraini [24], Kuwaiti [27], and Saudi women [26].

Women who were younger, were employed, were premenopausal, were single, had low income, had undergone chemotherapy or dissection and were highly educated seemed to have poorer body image. Therefore, although doing better on the physical side, younger women did worse on the emotional and body image scales. This is in line with the literature illustrating body image and sexuality issues as disturbing potential consequences of treatment for younger women with BC in particular [35]. Furthermore, monthly income was the only predictor that had a significant effect on global health/QoL given the other predictors in the regression model.

Our study indicated that Emirati women with BC experience problems in some of the QoL domains, and further research in this direction is recommended, especially in the emotional, sexual and side effect domains, to understand how better these domains could be evaluated and managed among Emirati women with special consideration of cultural sensitivity.

The study highlighted important aspects of the disease, such as side effects of the treatment and the need to explain them to patients along with ways of coping and adjustment. Furthermore, we recommend that special attention be given to women with metastatic history, as the impact on their QoL is substantial.

## 5. Strengths and Limitations of the Study

This study has several strengths, such as the random sampling method, the use of validated tools to measure the outcome, and the use of a standardized score for analysis

One important strength of our study is that it is the first study to investigate the QoL of Emirati menopausal women in the UAE.

Limitations of our study consist of the small sample size; this was explained by the gynecologists working in the primary health care centers that Emirati women do not consult them only in case of an emergency, and if they do, they only consult female doctors. Due to the limited number of participants in the study, it was unfeasible to categorize them by age group. Other limitations of our study include the descriptive design, absence of comparison group, recall bias. Choosing the interview instead of self−administered survey is a double edge sword. When the interview provided a consistent and clear description of the questions, the self−administered survey gives more space and freedom to answer sensitive questions.

## 6. Conclusions

Emirati women with BC were shown to have a good QoL compared to women in the region. As assessed by EORTC QLQ−C30 and QLQ−BR23, Emirati women with BC performed well on the functional scales but poorly on the symptom scale, scored the highest on social functioning and the lowest on emotional functioning scored the lowest. Sleep disturbance, fatigue, pain, hair loss and arm symptoms were the most bothersome symptoms. Factors associated with a decline in the domains of QoL included higher age, lower monthly income, and history of metastases, mastectomy, and lymph node dissection. The study contributes to a better understanding of the QoL of Emirati women with BC and the different factors that affect their wellbeing.

## Figures and Tables

**Table 1 ijerph-20-00570-t001:** Demographic data of the participants (N = 250).

Characteristic	Frequency	Percentage
Age		
≤50	107	42.8
>50	143	57.2
Marital Status		
Single	15	6.0
Married	187	74.8
Divorced	16	6.4
Widowed	32	12.8
Education Level		
Illiterate	45	18
Primary School	32	12.8
Preparatory School	23	9.2
Secondary School	56	22.4
University Graduate	94	37.6
Employment		
Yes	58	23.2
No	174	69.6
Retired	18	7.2
Menopausal Status		
Premenopause	26	10.4
Perimenopause	24	9.6
Postmenopause	153	61.2
Menopause due to surgery	47	18.8
Monthly Income		
<10,000 AED (<$2700)	40	16.0
10,000–20,000 AED ($2700–$5400)	117	46.8
20,000–30,000 AED ($5400–$8000)	73	29.2
>30,000 AED (>$8000)	20	8.0
Time Since Diagnosis		
Early Diagnosis (≤1 year)	46	18.4
Transitional Period (>1 and ≤5 years)	136	54.4
Long-term Survivors (>5 to ≤10 years)	68	27.2
Stage		
Stage 1	67	26.8
Stage 2	55	22.0
Stage 3	53	21.2
Stage 4	12	4.8
I don’t know	63	25.2
Metastasis		
Yes	29	11.6
No	215	86.0
I don’t know	6	2.4
Type of Treatment		
Chemotherapy	181	72.4%
Radiotherapy	146	58.4%
Hormonal therapy	138	55.2%
Lumpectomy	106	42.4%
Mastectomy	128	51.2%
Lymph node dissection	78	31.2%

**Table 2 ijerph-20-00570-t002:** Mean scores for all items on the QLQ-C30 and QLQ-BR23 (N = 250).

Variables	No. of Items	Mean (SD)	95% CI	N (%) (Scoring < 33.3) ^a^	N (%) (Scoring ≥ 66.7) ^b^
QLQ-C30					
Global health status/QoL	2	74.73 (18.25)	72.46–77.00	4 (1.6)	190 (76.0)
Functional scales ^b^					
Physical functioning	5	69.95 (25.13)	66.82–73.08	22 (8.8)	166 (66.4)
Role functioning	2	79.13 (28.36)	75.60–82.67	14 (5.6)	199 (79.6)
Emotional functioning	4	68.43 (30.02)	64.69–72.17	31 (12.4)	168 (67.2)
Cognitive functioning	2	74.93 (27.51)	71.51–78.36	17 (6.8)	190 (76.0)
Social functioning	2	82.33 (28.38)	78.80–85.87	16 (6.4)	209 (83.6)
Symptom scales ^c^					
Fatigue	3	38.18 (30.31)	34.40–41.95	105 (42.0)	58 (23.2)
Nausea and vomiting	2	18.80 (27.17)	15.42–22.18	168 (67.2)	26 (10.4)
Pain	2	29.13 (28.01)	25.64–32.62	128 (51.2)	41 (16.4)
Dyspnea	1	20.53 (29.66)	16.84–24.23	150 (60.0)	39 (15.6)
Sleep disturbance- Insomnia	1	47.87 (38.46)	43.08–52.66	70 (28.0)	114 (45.6)
Appetite loss	1	25.87 (32.96)	21.76–29.97	130 (52.0)	48 (19.2)
Constipation	1	26.93 (35.27)	22.54–31.33	136 (54.4)	56 (22.4)
Diarrhea	1	12.13 (26.54)	8.83–15.44	195 (78.0)	22 (8.8)
Financial impact	1	9.2 (23.13)	6.32–12.08	207 (82.8)	17 (6.8)
QLQ-BR23					
Functional scales ^b^					
Body image	4	80.30 (25.73)	77.09–83.51	17 (6.8)	204 (81.6)
Sexual functioning	2	86.07 (22.61)	83.25–88.88	5 (2.0)	224 (89.6)
Sexual enjoyment (N = 78)	1	63.68 (30.48)	56.80–70.53	6 (2.4)	54 (21.6)
Future perspective	1	50.80 (37.92)	46.08–55.52	60 (24.0)	121 (48.4)
Symptom scales ^c^					
Systemic side effect	7	31.98 (25.37)	28.82–35.14	134 (53.6)	31 (12.4)
Breast symptoms	4	26.93 (27.90)	23.46–30.41	154 (61.6)	33 (13.2)
Arm symptoms	3	33.73 (28.08)	30.23–37.23	117 (46.8)	48 (19.2)
Upset by hair loss	1	61.01 (37.35)	55.16–66.86	28 (11.2)	100 (40.0)

^a^ For the functional scales, women scoring <33.3% had problems; those scoring ≥66.7% had good functioning. For the symptom scales/symptoms, women scoring <33.3% had good functioning; those scoring ≥66.7% had problems. ^b^ For the functional scales, higher scores indicate better functioning. ^c^ For the symptom scales, higher scores indicate worse functioning.

**Table 3 ijerph-20-00570-t003:** Global health and functional scales on the QLQ-C30 by independent variables (N = 250) ^a^.

Characteristic	Global Health Status/Qol Mean (SD)	Functional Scales in QLQ-C30 ^b^				
		Physical Functioning Mean (SD)	Role Functioning Mean (SD)	Emotional Functioning Mean (SD)	Cognitive Functioning Mean (SD)	Social Functioning Mean (SD)
Age						
≤50 years	74.84 (18.06)	70.65 (24.84)	77.88 (30.18)	58.26 (31.02)	69.47 (27.99)	72.43 (30.47)
>50 years	74.65 (18.45)	69.42 (25.42)	80.07 (26.99)	76.05 (26.94)	79.02 (26.52)	89.74 (24.30)
*p*	0.969	0.725	0.638	<0.0001	0.002	<0.0001
Time since diagnosis						
Early Diagnosis	75.00 (18.00)	63.79 (27.78)	69.20 (32.39)	65.03 (28.74)	73.91 (28.47)	77.90 (26.07)
Transitional Period	75.92 (18.14)	71.23 (25.03)	81.62 (26.86)	65.01 (30.19)	74.02 (27.57)	81.99 (28.84)
Long-term Survivors	72.18 (18.64)	71.57 (23.11)	80.88 (27.36)	77.57 (29.02)	77.45 (26.98)	86.03 (28.88)
*p*	0.417	0.257	0.043	0.002	0.637	0.037
Marital Status						
Single	75.00 (19.16)	72.89 (25.38)	87.78 (22.24)	51.11 (34.63)	65.56 (37.52)	76.67 (28.73)
Married	74.20 (17.86)	70.05 (24.38)	79.23 (28.47)	67.20 (29.88)	75.94 (26.27)	81.11 (29.37)
Divorced	82.81 (16.52)	72.50 (25.40)	82.29 (29.48)	66.67 (31.18)	72.92 (30.35)	83.33 (29.81)
Widowed	73.70 (20.64)	66.67 (29.77)	72.91 (29.56)	84.63 (21.07)	74.48 (28.39)	91.67 (19.40)
*p*	0.350	0.894	0.279	0.001	0.817	0.108
Education Level						
Illiterate	73.33 (18.00)	63.85 (28.37)	72.59 (29.34)	79.26 (26.45)	78.89 (26.45)	92.96 (18.63)
Primary School	75.26 (17.77)	74.79 (20.02)	88.02 (24.40)	78.91 (27.92)	83.85 (26.26)	91.67 (21.59)
Preparatory School	73.91 (23.74)	65.51 (24.10)	78.26 (24.33)	77.90 (26.78)	77.54 (22.81)	92.75 (19.35)
Secondary School	71.43 (17.61)	67.50 (25.11)	72.92 (31.87)	63.54 (34.69)	70.24 (27.84)	73.21 (32.52)
University Graduate	77.39 (17.35)	73.76 (24.84)	83.16 (26.83	60.28 (27.26)	72.16 (28.75)	76.95 (30.54)
*p*	0.355	0.130	0.012	<0.0001	0.057	<0.0001
Employment						
Yes	75.86 (16.72)	74.37 (24.24)	84.20 (25.44)	60.63 (26.66)	75.29 (25.60)	76.15 (29.64)
No	73.75 (18.32)	67.93 (25.44)	77.78 (28.01)	70.88 (30.83)	74.62 (27.74)	84.39 (27.65)
Retired	80.56 (21.77)	75.19 (23.63)	75.93 (38.87)	69.91 (29.59)	76.85 (32.41)	82.41 (29.96)
*p*	0.132	0.113	0.234	0.014	0.819	0.028
Monthly Income						
<$2700	69.58 (21.89)	64.33 (24.56)	72.08 (29.81)	61.46 (32.78)	69.58 (33.52)	75.42 (33.97)
$2700–$5400	72.93 (17.91)	68.26 (24.11)	79.06 (30.02)	70.51 (29.04)	74.93 (25.49)	83.05 (27.42)
$5400-$8000	78.65 (15.96)	77.44 (23.87)	80.82 (26.45)	72.26 (30.71)	79.00 (26.21)	89.73 (22.33)
>$8000	81.25 (16.64)	63.67 (31.45)	87.50 (19.40)	56.25 (23.24)	70.83 (30.05)	65.00 (32.84)
*p*	0.018	0.007	0.232	0.018	0.399	<0.001
Physical Activity						
Yes	79.08 (17.76)	77.42 (22.52)	85.88 (21.05)	66.67 (29.48)	77.38 (25.13)	75.85 (32.12)
No	71.93 (18.06)	65.13 (25.61)	74.78 (31.52)	69.57 (30.41)	73.36 (29.92)	88.51 (24.92)
*p*	0.004	<0.0001	0.013	0.331	0.382	0.003
Menopausal Status						
Premenopause	71.47 (20.97)	69.74 (20.76)	75.64 (30.63)	59.94 (29.54)	73.72 (29.50)	72.44 (31.25)
Perimenopause	78.13 (20.09)	62.50 (35.16)	68.06 (38.04)	72.92 (27.39)	75.69 (25.05)	72.22 (32.10)
Postmenopause	74.40 (17.73)	70.59 (24.94)	82.68 (25.64)	73.37 (28.92)	78.10 (26.18)	88.67 (24.97)
Menopause due to Surgery	75.89 (17.57)	71.77 (21.86)	75.18 (28.63)	54.79 (30.73)	64.89 (30.14)	72.34 (30.15)
*p*	0.555	0.883	0.111	<0.0001	0.042	<0.0001
Children						
Yes	74.16(18.20)	69.89(24.64)	78.16(28.71)	69.18(29.73)	75.57(26.96)	82.42(28.54)
No	78.76(18.36)	70.32(28.79)	86.02(25.13)	63.17(32.04)	70.43(31.24)	81.72(27.67)
*p* value	0.227	0.618	0.091	0.271	0.398	0.688
Stage						
Stage 1	78.48 (16.10)	77.21 (22.24)	86.57 (23.97)	66.04 (29.49)	79.10 (24.85)	86.82 (25.71)
Stage 2	73.03 (17.27)	65.33 (23.36)	78.48 (28.09)	65.15 (28.42)	69.70 (26.66)	72.73 (33.39)
Stage 3	70.28 (18.89)	70.31 (21.35)	80.82 (24.55)	64.62 (31.90)	69.50 (26.90)	77.36 (27.56)
Stage 4	73.61 (21.57)	67.22 (31.33)	75.00 (35.18)	66.67 (30.77)	73.61 (33.68)	86.11 (30.01)
*p*	0.069	0.025	0.168	0.981	0.093	0.023
Metastases						
Yes	64.66 (17.49)	62.76 (22.50)	77.59 (27.92)	72.13 (31.44)	76.44 (29.72)	84.48 (29.19)
No	76.36 (17.70)	71.53 (25.046)	79.77 (28.33)	68.06 (29.83)	75.04 (27.10)	82.56 (28.09)
*p*	0.001	0.027	0.599	0.347	0.657	0.430
Chemotherapy						
Yes	73.30 (19.02)	68.69 (24.72)	79.37 (27.97)	66.16 (30.97)	71.92 (27.97)	79.37 (30.09)
No	78.50 (15.55)	73.24 (26.06)	78.50 (29.58)	74.40 (26.67)	82.85 (24.75)	90.10 (21.64)
*p*	0.045	0.110	0.929	0.069	0.002	0.006
Radiotherapy						
Yes	74.03 (18.30)	70.37 (22.95)	80.37 (26.67)	67.12 (28.88)	73.86 (28.22)	78.77 (30.23)
No	75.72 (18.22)	69.36 (28.02)	77.40 (30.63)	70.27 (31.60)	76.44 (26.54)	87.34 (24.85)
*p*	0.438	0.701	0.759	0.190	0.563	0.006
Lumpectomy						
Yes	73.90 (19.69)	68.49 (26.26)	80.19 (26.64)	68.40 (28.26)	73.90 (27.12)	78.14 (31.32)
No	75.35 (17.15)	71.02 (24.30)	78.36 (29.64)	68.46 (31.35)	75.69 (27.87)	85.42 (25.69)
*p*	0.504	0.564	0.935	0.605	0.536	0.030
Mastectomy						
Yes	75.00 (17.56)	70.42 (23.42)	79.43 (28.56)	66.21 (31.41)	75.78 (26.61)	83.46 (26.08)
No	74.45 (19.01)	69.45 (26.90)	78.83 (28.27)	70.77 (28.43)	74.04 (28.51)	81.15 (30.68)
*p*	0.659	0.801	0.606	0.360	0.828	0.792
Lymph Node Dissection						
Yes	72.76 (18.59)	68.72 (23.40)	82.69 (27.84)	57.69 (31.66)	66.03 (30.08)	69.87 (32.68)
No	75.63 (18.07)	70.50 (25.93)	77.52 (28.53)	73.30 (28.01)	78.97 (25.34)	87.98 (24.27)
*p*	0.246	0.350	0.090	<0.0001	0.001	<0.0001

^a^*p* based on Kruskal–Wallis or Mann–Whitney tests. ^b^ For the functional scales, higher scores indicate better functioning.

**Table 4 ijerph-20-00570-t004:** Functional and symptom scales on the QLQ-BR23 by independent variables (N = 250) ^a^.

	Functional Scales in BR 23 ^b^				Symptom Scale in BR 23 ^c^			
Characteristic	Body Image Mean (SD)	Sexual Functioning Mean (SD)	Sexual Enjoyment Mean (SD)	Future Perspective Mean (SD)	Systemic-Therapy Side Effect Mean (SD)	Breast Symptoms Mean (SD)	Arm Symptoms Mean (SD)	Upset by Hair LossMean (SD)
Age								
≤50 years	72.98 (27.99)	74.92 (26.93)	58.18 (30.91)	42.68 (37.98)	38.63 (26.00)	34.11 (27.29)	39.56 (26.00)	67.57 (34.88)
>50 years	85.78 (22.48)	94.41 (13.84)	76.81 (25.49)	56.88 (36.86)	27.01 (23.79)	21.56 (27.29)	29.37 (28.87)	55.29 (38.68)
*p*	<0.0001	<0.0001	0.015	0.003	<0.0001	<0.0001	0.001	0.047
Time Since Diagnosis								
Early Diagnosis	76.45 (25.36)	83.33 (22.77)	69.70 (27.04)	46.38 (40.05)	39.03 (28.17)	36.59 (32.23)	39.86 (27.17)	67.68 (32.79)
Transitional Period	80.27 (24.70)	82.84 (25.25)	59.57 (32.55)	48.53 (37.81)	31.30 (25.27)	27.51 (26.69)	33.74 (27.30)	57.92 (39.24)
Long-term Survivors	82.97 (27.97)	94.36 (13.07)	70.37 (26.06)	58.33 (36.14)	28.57 (22.93)	19.24 (25.20)	29.58 (29.84)	61.59 (37.16)
*p*	0.070	0.001	0.339	0.147	0.151	0.004	0.092	0.526
Marital Status								
Single	75.00 (29.55)	94.44 (10.29)	-	31.11 (29.46)	37.46 (26.38)	38.89 (25.33)	37.78 (26.49)	43.59 (34.39)
Married	78.39 (27.02)	82.35 (24.70)	62.16 (30.38)	50.45 (38.70)	32.31 (24.93)	26.43 (27.04)	34.11 (27.48)	61.98 (37.09)
Divorced	82.81 (19.60)	95.83 (9.62)	88.89 (19.25)	60.42 (38.91)	27.98 (30.56)	27.60 (31.87)	34.72 (31.66)	83.33 (27.89)
Widowed	92.70 (12.66)	98.96 (4.10)	100 (−)	57.29 (34.11)	29.46 (25.38)	23.96 (31.66)	29.17 (31.14)	59.65 (40.94)
*p*	0.029	<0.0001	0.131	0.113	0.546	0.165	0.576	0.146
Education Level								
Illiterate	91.48 (14.38)	95.93 (10.75)	66.67 (0.00)	66.67 (32.57)	23.17 (20.97)	20.74 (24.98)	31.11 (29.64)	56.52 (36.84)
Primary School	84.11 (25.95)	93.75 (22.70)	22.22 (38.49)	61.46 (33.98)	26.04 (26.14)	16.93 (22.05)	29.17 (24.07)	62.50 (43.67)
Preparatory School	85.87 (14.31)	82.61 (29.08)	74.07 (43.39)	57.97 (40.47)	33.95 (27.10)	28.26 (28.95)	31.88 (23.52)	50.98 (42.68)
Secondary School	80.06 (28.32)	83.93 (24.20)	55.56 (24.34)	45.83 (38.44)	31.97 (22.97)	28.57 (29.81)	32.54 (27.94)	62.63 (39.75)
University Graduate	72.43 (28.13)	80.85 (22.26)	68.22 (28.13)	40.78 (37.60)	37.74 (26.79)	32.00 (28.67)	37.71 (29.70)	63.81 (33.93)
*p*	<0.0001	<0.0001	0.070	0.001	0.020	0.037	0.560	0.758
Employment								
Yes	70.55 (29.79)	73.85 (21.88)	65.74 (29.26)	39.66 (38.21)	37.44 (30.11)	38.51 (32.25)	38.51 (28.63)	61.11 (32.02)
No	84.05 (23.41)	90.42 (21.45)	56.86 (31.28)	56.13 (37.04)	30.43 (23.71)	24.28 (26.24)	32.06 (28.12)	61.68 (39.60)
Retired	75.46 (25.48)	83.33 (22.14)	83.33 (25.20)	35.19 (35.19)	29.37 (23.11)	15.28 (15.19)	34.57 (25.53)	53.33 (35.83)
*p*	0.001	<0.0001	0.079	0.003	0.404	0.005	0.277	0.683
Monthly Income								
<10,000 AED	68.75 (28.42)	77.50 (32.81)	35.71 (30.56)	35.00 (39.19)	42.62 (27.34)	42.92 (32.93)	45.56 (32.88)	66.67 (40.57)
10,000–20,000 AED	81.20 (26.94)	87.04 (20.31)	63.89 (26.87)	54.99 (37.73)	29.30 (23.98)	23.01 (24.99)	34.76 (25.79)	54.79 (37.83)
20,000–30,000 AED	84.93 (21.41)	91.10 (17.59)	72.55 (24.25)	52.97 (35.50)	26.29 (22.28)	18.84 (21.56)	20.24 (22.09)	66.67 (34.16)
>30,000 AED	81.25 (21.94)	79.17 (22.21)	84.85 (27.34)	50.00 (39.74)	47.14 (29.35)	47.50 (32.68)	53.33 (28.29)	64.71 (36.27)
*p*	0.011	0.021	0.001	0.032	0.001	<0.0001	<0.0001	0.267
Physical Activity								
Yes	74.32 (29.13)	82.48 (23.86)	65.74 (34.26)	48.30 (38.03)	32.75 (28.15)	29.51 (27.66)	36.28 (28.39)	64.33 (38.25)
No	84.16 (22.55)	88.38 (21.53)	61.90 (27.12)	52.41 (37.90)	31.49 (23.49)	25.27 (28.02)	32.09 (27.86)	59.15 (36.91)
*p*	0.005	0.014	0.469	0.406	0.945	0.149	0.219	0.345
Menopausal Status								
Premenopause	70.51 (28.99)	69.23 (35.49)	45.24 (36.06)	34.62 (38.27)	37.91 (23.15)	41.03 (23.80)	38.89 (22.93)	61.90 (38.42)
Perimenopause	81.25 (23.47)	79.17 (27.03)	57.58 (26.21)	59.72 (36.75)	34.13 (30.07)	30.21 (30.97)	40.74 (34.31)	69.05 (38.04)
Postmenopause	84.20 (24.22)	92.37 (15.65)	72.22 (27.80)	56.64 (37.48)	28.73 (24.83)	21.79 (26.22)	29.77 (27.59)	58.42 (38.59)
Menopause due to Surgery	72.52 (27.30)	78.37 (23.03)	66.67 (28.43)	36.17 (33.93)	38.20 (24.63)	34.22 (29.91)	40.19 (27.27)	64.52 (33.26)
*p*	0.001	<0.0001	0.076	0.001	0.046	<0.0001	0.034	0.724
Children								
Yes	80.48(25.90)	85.31(23.32)	63.01(29.69)	51.14(38.35)	31.72(24.72)	26.75(27.89)	33.59(28.37)	63.02(37.20)
No	79.03(24.90)	91.40(16.03)	73.33(43.46)	48.39(35.32)	33.79(29.95)	28.23(28.36)	34.77(26.41)	48.48(36.70)
*p*	0.583	0.264	0.312	0.723	0.919	0.900	0.676	0.082
Stage								
Stage 1	83.33 (22.89)	83.58 (24.36)	59.09 (28.97)	53.23 (38.95)	23.67 (23.45)	24.01 (26.21)	28.36 (25.17)	50.51 (39.19)
Stage 2	76.67 (22.36)	83.94 (25.45)	63.16 (31.22)	42.42 (36.55)	43.46 (22.09)	30.30 (27.38)	38.99 (28.84)	71.32 (25.80)
Stage 3	69.50 (33.45)	79.56 (25.87)	58.67 (32.32)	49.06 (38.46)	39.08 (28.76)	34.12 (30.10)	40.46 (31.13)	63.06 (39.11)
Stage 4	88.19 (13.97)	90.28 (13.22)	80.00 (18.26)	38.89 (39.78)	36.90 (27.78)	41.67 (37.61)	38.89 (27.83)	70.00 (33.15)
*p*	0.044	0.571	0.524	0.369	<0.0001	0.179	0.085	0.144
Metastases								
Yes	81.61 (27.85)	85.63 (20.76)	53.33 (28.11)	39.08 (36.81)	40.56 (23.69)	34.20 (31.52)	41.38 (31.97)	66.67 (33.33)
No	80.00 (25.69)	85.81 (23.10)	65.20 (30.71)	52.25 (37.91)	30.32 (25.16)	25.16 (26.58)	32.04 (26.90)	59.38 (38.32)
*p*	0.452	0.767	0.219	0.079	0.028	0.186	0.125	0.417
Chemotherapy								
Yes	77.44 (27.91)	85.36 (22.61)	66.12 (30.73)	48.43 (36.76)	35.12 (26.17)	25.92 (26.44)	36.65 (28.01)	62.27 (38.27)
No	87.80 (16.82)	87.92 (22.67)	54.90 (28.73)	57.00 (40.46)	23.74 (21.19)	29.59 (31.46)	26.09 (27.00)	55.56 (33.14)
*p*	0.031	0.196	0.130	0.116	0.002	0.678	0.004	0.261
Radiotherapy								
Yes	76.66 (27.81)	85.16 (22.16)	71.70 (30.24)	46.12 (36.99)	35.84 (26.23)	29.85 (28.12)	39.35 (27.50)	62.96 (36.23)
No	85.42 (21.59)	87.34 (23.28)	46.67 (23.57)	57.37 (38.43)	26.56 (23.16)	22.84 (27.18)	25.85 (27.10)	57.78 (39.23)
*p*	0.009	0.198	<0.0001	0.021	0.006	0.026	<0.0001	0.456
Lumpectomy								
Yes	79.40 (24.81)	86.48 (21.96)	64.58 (31.61)	51.57 (37.97)	33.74 (26.63)	30.82 (28.73)	34.80 (27.90)	62.63 (34.85)
No	80.96 (26.46)	85.76 (23.15)	63.04 (30.00)	50.23 (38.02)	30.69 (24.42)	24.07 (27.01)	32.95 (28.29)	59.86 (39.18)
*p*	0.339	0.892	0.752	0.784	0.444	0.035	0.578	0.797
Mastectomy								
Yes	76.37 (28.23)	85.03 (24.20)	59.52 (31.70)	46.35 (37.47)	31.92 (24.63)	25.65 (26.96)	37.33 (27.98)	63.41 (37.99)
No	84.43 (22.19)	87.16 (20.86)	68.52 (28.66)	55.46 (38.00)	32.05 (26.23)	28.28 (28.90)	29.96 (27.81)	58.44 (36.74)
*p*	0.024	0.675	0.197	0.058	0.888	0.577	0.028	0.338
Lymph Node Dissection								
Yes	71.37 (30.26)	83.76 (24.02)	67.86 (33.31)	47.01 (37.00)	42.37 (25.27)	32.37 (27.88)	42.02 (28.41)	64.94 (35.55)
No	84.35 (22.34)	87.11 (21.93)	61.33 (28.86)	52.52 (38.32)	27.27 (24.05)	24.47 (27.63)	29.97 (27.19)	58.75 (38.34)
*p*	<0.0001	0.218	0.256	0.296	<0.0001	0.017	0.002	0.358

^a^*p* based on Kruskal–Wallis or Mann–Whitney tests. ^b^ For the functional scales, higher scores indicate better functioning. ^c^ For the symptom scales, higher scores indicate worse functioning. 1$ = 3.67 AED.

**Table 5 ijerph-20-00570-t005:** Linear regression model with parameter estimates for the QLQ functional scales.

Variable	Global Health/QoL Score		Physical Functioning		Role Functioning		Emotional Functioning		Cognitive Functioning		Social Functioning	
	*β*	*p*	*β*	*p*	*β*	*p*	*β*	*p*	*β*	*p*	*β*	*p*
Constant	80.704	<0.001	74.274	<0.001	73.346	<0.001	65.078	<0.001	66.378	<0.001	87.544	<0.001
Age > 50	−0.079	0.412	−0.147	0.134	0.050	0.615	0.262	0.004	0.115	0.219	0.074	0.419
Married	−0.063	0.409	0.026	0.732	0.025	0.753	0.047	0.510	0.124	0.093	0.018	0.808
Education	−0.003	0.972	0.033	0.708	−0.079	0.383	−0.220	0.008	−0.125	0.142	−0.197	0.019
Employment	0.008	0.921	0.013	0.873	0.138	0.099	0.021	0.776	0.152	0.051	−0.002	0.979
High income	0.242	0.002	0.127	0.099	0.160	0.043	0.031	0.660	0.062	0.403	0.059	0.416
Menopause	0.042	0.619	0.100	0.244	0.113	0.199	−0.077	0.336	−0.022	0.789	0.098	0.227
Advanced stage	−0.052	0.510	0.024	0.766	−0.039	0.635	0.034	0.649	−0.084	0.276	0.027	0.721
Long−time survivors	−0.033	0.677	0.070	0.381	0.044	0.588	0.074	0.316	0.071	0.352	−0.038	0.613
Metastases	−0.140	0.079	−0.141	0.081	−0.013	0.874	0.046	0.537	0.168	0.031	0.083	0.272
Chemotherapy	−0.170	0.035	−0.164	0.047	−0.021	0.799	−0.035	0.648	−0.150	0.056	−0.151	0.051
Radiotherapy	−0.021	0.805	−0.034	0.696	−0.087	0.323	0.049	0.542	0.064	0.436	−0.104	0.199
Lumpectomy	0.064	0.550	0.043	0.689	0.121	0.272	0.113	0.259	0.150	0.149	0.097	0.339
Mastectomy	0.028	0.781	0.022	0.833	0.071	0.501	0.034	0.723	0.194	0.050	0.157	0.107
Lymph node dissection	−0.063	0.469	−0.174	0.050	0.038	0.674	−0.179	0.029	−0.203	0.017	−0.184	0.027
Hormonal therapy	−0.030	0.709	0.040	0.631	−0.098	0.244	−0.013	0.861	−0.052	0.507	−0.050	0.517
No = 0												
Yes = 1												
R^2^	0.063		0.033		0.007		0.169		0.119		0.147	
*p*	0.036		0.148		0.546		<0.001		0.001		<0.001	

*β*: Standardized Coefficient Beta. *p*: Significance.

## Data Availability

The datasets analyzed during the current study are available from the corresponding author on reasonable request.

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
