# Peer review of "Quality of Life of Emirati Women with Breast Cancer"

_ijerph, 2022, doi:10.3390/ijerph20010570_

Round 1

Reviewer 1 Report

The manuscript entitled ‘Quality of life of Emirati women with breast cancer’ presents important issue, however some corrections are needed. 

-        Line 12 – ‘250 Emirati women’ (The sample size was determined based on a power of 85% and a significant 90 level of 5%). – How was the sample size established? It should be justify.

-        Lines 69-70 – ‘September 2020 to 69 April 2021’ – how COVID-19 pandemic could influence the health care and the obtained results?

-        It is unclear why ‘One-way analysis of variance’ was conducted?

-        There are high share of ‘University Graduate‘ participants? How it could influence the obtained results?  

-        There are very high share of ‘unemployment ‘ participants? How it could influence the obtained results?  

-        Table 1 – ‘Monthly Income’ – please add also information in dollars or euros (for international readers)

-        ‘20000-3–0000 AED’ – some typo

-        ‘ Time Since Diagnosis’ – the period should be define (‘Early Diagnosis’ – what does it means?)

-        Discussion section – please add some more intranational context (not only BC survivors women form Bahrain, Kuwait and Saudi Arabia)

-        In this manuscript Authors presented the information associated women with breast cancer survivors but in the present form it has national impact. The situation/ comparison with other countries should be presented for international readers.

Author Response

Comments and Suggestions for Authors

-        Line 12 – ‘250 Emirati women’ (The sample size was determined based on a power of 85% and a significant 90 level of 5%). – How was the sample size established? It should be justify.

Answer:

This was mentioned in Line 91, we changed no this to Line 73 and completed the information about the sample size computation.

The sample size was calculated in relation with the hypothesis that QoL of Emirati women with breast cancer is similar to what was found in a previous study in the UAE [4] (including 87 women not all Emirati)

This above mentioned study yielded a mean ± SD QoL score of 74.6 ± 18.2. By setting a 95% confidence interval within three standard deviations (ME) of the population mean using the following:

A total of 170 participants was decided allowing for a 20% non-response rate.

We also stated that the participation rates were 100% and around 66% from Dubai and Tawam hospitals respectively

We believe Covid-19 situation helped us getting a high number of participants as explained below, number of cancer patients follow-ups in these 2 major hospitals increased during covid and especially after the lockdown because  the UAE government had suspended all funding for offshore medical treatment, including oncology care. Therefore this helped in having a higher number of participants that was never achieved in a breast cancer study in the UAE.

Indeed this believe was supported by recent research in the UAE: https://www.ncbi.nlm.nih.gov/pmc/articles/PMC9176180/

-        Lines 69-70 – ‘September 2020 to 69 April 2021’ – how COVID-19 pandemic could influence the health care and the obtained results?

Answer:

The study was approved just before Covid and therefore questions like those mentioned above were not included in the study. We were not able to add any question related to Covid-19 as it was against ethical approvals obtained from all entities.

It is really worth pursuing as a future study, we will definitely consider this in the near future.

I would like to open a brackets here just to share our experience about running the study under Covid-19 and what I noticed from following up closely data collection with the nurses.

The lockdown in the UAE was from March 8, 2020, to almost end of May 2022, and partial lockdown after that, but visits to hospital for medical reasons or even for research could be easily obtained. We worked on ethical clearance during this period of time.

The interviews were run by nurses during follow up of Emirati women with BC in the day care clinics. After they agreed to take the survey and signed consent form, the nurses sat with them, asked them questions one by one and filled in the survey form.

Tawam hospital is only a cancer hospital and therefore it was open to all cancer patients all time with no restrictions at all.  Dubai hospital was a bit more restricted as they had a service for Covid, that was in a separate building but my visits to nurses were only in the cancer building, so I had no issue at all, just the fear of Covid-19 itself. Same for patients visits, they came to their follow up, and if they canceled, they always reschedule.

 It was different for patients who used to follow up their treatment outside the country, because the UAE government had suspended all funding for offshore medical treatment, including oncology care, to avoid placing patients and companions at a higher risk of acquiring the virus. Therefore, the number of visits by Breast cancer patients to both hospitals increased as it was the only option available to those patients who are not registered in any of the hospitals.

The only constraint from Covid-19 itself, that hospitals administration asked us to stop using paper forms and to find an alternative. We provided each nurse with an iPad that had the survey on a form and data was collected on the iPad directly.

To summarize, we believe that Covid didn’t affect our results, nurses were trained by a BC specialist in how to approach women and ask each question to get as much as possible accurate answers. All the extra comments given by women during the interviews were about finally being able to go out after the lockdown.

-        It is unclear why ‘One-way analysis of variance’ was conducted?

Answer:

Not only One-way analysis of variance (ANOVA) that was used in the whole study, but as mentioned in Lines 179-181 the use of t-test as well.

For example Education level was a variable with more than 3 categories and therefore ANOVA was used in this case. Binary Variables were only used for the multiple linear regression models.

-        There are high share of ‘University Graduate‘ participants? How it could influence the obtained results?  

-        There are very high share of ‘unemployment ‘ participants? How it could influence the obtained results?  

Answer:

It is worth mentioning that the 37.6% of university graduate is not high comparing to the total of the two categories (18% illiterate & 12.8% Primary School) as those are considered not having any education at all.

The UAE government has laid great emphasis on women’s education and empowerment. More than 70% of students enrolled in higher education institutions comprise of women. Despite the high literacy rate only approximately 34% of the UAE workforce comprises of women. This low level of employment could be reflective of sociocultural norms of the society, that are beginning to recognize the importance of women’s education but are only gradually accepting women in the workforce. (Reference below)

Surprisingly, there were no significant difference in Health-Related QoL among Emirati women with respect to their education level or employment status. Nevertheless, we could consider the possibility that breast cancer diagnosis and treatment was higher among more educated women.

For Love, Money and Status, or Personal Growth? A Survey of Young Emirati Women’s Educational Aspirations. (2020). Gulf Education and Social Policy Review1(1), 73–90. https://doi.org/10.18502/gespr.v1i1.7470

-        Table 1 – ‘Monthly Income’ – please add also information in dollars or euros (for international readers)

Answer: We added the information in $ as well

-        ‘20000-3–0000 AED’ – some typo

-        ‘ Time Since Diagnosis’ – the period should be define (‘Early Diagnosis’ – what does it means?)

Answer:

Typos correct, thank you. Also, diagnosis Cycles were defined in Lines 104-106, we added them to the table.

-        Discussion section – please add some more intranational context (not only BC survivors women form Bahrain, Kuwait and Saudi Arabia)

-        In this manuscript Authors presented the information associated women with breast cancer survivors but in the present form it has national impact. The situation/ comparison with other countries should be presented for international readers.

Answer:

We completed the comparison part using the below added references:

[28]   Global quality of life in breast cancer: systematic review and meta-analysis

  1. Javan Biparva, S. Raoofi, S. Rafiei, F. Pashazadeh Kan, M. Kazerooni, F. Bagheribayati, et al.

BMJ Supportive & Palliative Care 2022 Pages bmjspcare-2022-003642

DOI: 10.1136/bmjspcare-2022-003642

[29]     Mokhtari-Hessari, P., Montazeri, A. Health-related quality of life in breast cancer patients: review of reviews from 2008 to 2018. Health Qual Life Outcomes 18, 338 (2020). https://doi.org/10.1186/s12955-020-01591-x

[30]     Haddou Rahou B, El Rhazi K, Ouasmani F, Nejjari C, Bekkali R, Montazeri A, Mesfioui A. Quality of life in Arab women with breast cancer: a review of the literature. Health Qual Life Outcomes. 2016 Apr 27;14:64. doi: 10.1186/s12955-016-0468-9. PMID: 27117705; PMCID: PMC4847355.

[33]     Dupont P. In the Arab Bedroom: The Sex Life of Arabs. Facts Views Vis Obgyn. 2016 Dec;8(4):237-242. PMID: 28210484; PMCID: PMC5303702.

Reviewer 2 Report

I was pleased to read the manuscript entitled "Quality of life of Emirati women with breast cancer" and to review it.

The study was focused on the quality of life (QoL) among Emirati women with breast cancer (BC). From a scientific point of view, the article did not reveal particularly new regularities, but it is interesting to read because it is one of the few studies that was conducted in the selected population.

The article is written in a typical format. The rationale of the study is well described. Nevertheless, while reading the article several questions arose and inaccuracies were noticed, which I recommend to fix not only personally for the reviewer but probably to readers too.

1. Abstract: Specify the purpose of the study: "The study aimed to examine ...... and its relationships with their sociodemographic characteristics and ..." In fact, this study investigated the relationship of QoL not only with sociodemographic factors, but also with clinical factors, which are shown in the conclusions (e.g. "history of metastases, mastectomy, and lymph node dissection"). The aim of the study presented in the abstract must reflect the aim of the study stated in the Introduction. It must also be repeated at the beginning of the Discussion.

2. Sampling Method (lines 91-92): "The sample size was determined based on a power of 85% and a significant 90 level of 5%". In relation to what hypothesis was the sample size calculated? If it was calculated, what was the sample size?

3. Results, data presentation: In Statistical analysis it was stated "The  scoring algorithm involves first computing the average of the item responses and second transforming the score to a value on a 0-100% scale." So, when presenting the results (for example, "mean score of the 250 participants was 74.73 (SD ±18.25)"), you must indicate that these values are not the absolute sum of scores, but the relative sum of scores in a scale of 0-100%.

4. Statistical analysis: It is useful to provide data on the distributions of the dependent variables. They are obviously not normal. As a result, the use of linear regression is limited. The use of this model is also limited by the fact that all independent variables are binary variables. Thus, Poisson regression and better yet logistic regression would be more appropriate for this study.

5. Lines 417-422 and 424-433: It is advisable to avoid duplication.

6. Appendix: Numeric values should appear in line X=1. The format of Table 5 can also be applied.

7. Table 1: Remove N and % in line "Menopausal Status". Information about other variables should also be presented in this table.

Thank you for considering my opinion. I encourage authors to keep on working to improve the manuscript.

Author Response

Comments and Suggestions for Authors

I was pleased to read the manuscript entitled "Quality of life of Emirati women with breast cancer" and to review it.

The study was focused on the quality of life (QoL) among Emirati women with breast cancer (BC). From a scientific point of view, the article did not reveal particularly new regularities, but it is interesting to read because it is one of the few studies that was conducted in the selected population.

The article is written in a typical format. The rationale of the study is well described. Nevertheless, while reading the article several questions arose and inaccuracies were noticed, which I recommend to fix not only personally for the reviewer but probably to readers too.

Answer:

We really appreciate the feedback; we believe in the importance of this study for the community. We see already an impact, first on the nurses who were trained to run the survey and who expressed their impression that they never thought about some questions and aspects that can really tell them about women quality of life.

More important, when we shared the draft of the results with breast cancer heads of clinics, we received great feedback, one of them, I quote: “I believe you have highlighted a very important issue which is often overlooked. As Physicians we are more focused on disease control and outcomes however impact on Quality of life especially in long term survivors is very important.”

The UAE is an emerging country and there is a big gab in knowledge about their women QoL, therefore we hope this will bring answers to authorities and decision makers for better health care and policies related to women.

  1. Abstract: Specify the purpose of the study: "The study aimed to examine ...... and its relationships with their sociodemographic characteristics and ..." In fact, this study investigated the relationship of QoL not only with sociodemographic factors, but also with clinical factors, which are shown in the conclusions (e.g. "history of metastases, mastectomy, and lymph node dissection"). The aim of the study presented in the abstract must reflect the aim of the study stated in the Introduction. It must also be repeated at the beginning of the Discussion.

Answer:

We made the change as requested. It did make sense now that this is added to the abstract and beginning of the discussion section.

  1. Sampling Method (lines 91-92): "The sample size was determined based on a power of 85% and a significant 90 level of 5%". In relation to what hypothesis was the sample size calculated? If it was calculated, what was the sample size?

Answer:

The sample size was calculated in relation with the hypothesis that QoL of Emirati women with breast cancer is similar to what was found in a previous study in the UAE [4] (including 87 women not all Emirati)

This above mentioned study yielded a mean ± SD QoL score of 74.6 ± 18.2. By setting a 95% confidence interval within three standard deviations (ME) of the population mean using the following:

A total of 170 participants was decided allowing for a 20% non-response rate.

We also stated that the participation rates were 100% and around 66% from Dubai and Tawam hospitals respectively.

We believe Covid-19 situation helped us getting a high number of participants, the number of cancer patients visits to the 2 major hospitals in the UAE increased during Covid-19 and especially after the lockdown because  the UAE government had suspended all funding for offshore medical treatment, including oncology care. Therefore this helped in having a higher number of participants that was never achieved in a breast cancer study in the UAE.

Indeed this believe was supported by recent research in the UAE: https://www.ncbi.nlm.nih.gov/pmc/articles/PMC9176180/

  1. Results, data presentation: In Statistical analysis it was stated "The  scoring algorithm involves first computing the average of the item responses and second transforming the score to a value on a 0-100% scale." So, when presenting the results (for example, "mean score of the 250 participants was 74.73 (SD ±18.25)"), you must indicate that these values are not the absolute sum of scores, but the relative sum of scores in a scale of 0-100%.

Answer:

Thank you for noticing this. We added the following to the end of Line 205.

 It is important to note that all these reported values are not the absolute sum of scores, but the relative sum of scores in a scale of 0-100% as recommended by the EORTC [22, 23].

  1. Statistical analysis: It is useful to provide data on the distributions of the dependent variables. They are obviously not normal. As a result, the use of linear regression is limited. The use of this model is also limited by the fact that all independent variables are binary variables. Thus, Poisson regression and better yet logistic regression would be more appropriate for this study.

Answer:

Our dependent variables are mainly Health-Related QoL and all functions and Symptoms Scales that are all continuous, while independent variables are binary which is OK for multiple linear regressions. We didn’t use any of the binary variable ad dependent variable to justify the use of a Poisson or a logistic regression

We didn’t include all details about assessing normality, we mentioned briefly that linear regression assumption were checked in the methodology section, but we are adding them here for clarity.

Indeed, there were lots of assumptions associated with these multiple linear regressions that were all checked.

  1. there isn't too strong of a relationship between the predictor and dependent variables, the correlation or the r values were closer to zero than minus .7 and we concluded that there isn't a problem with no multicollinearity. We also checked tolerance (above .1) and VIF values (below 10)
  2. values of the residuals are independent. For this we checked the Durbin Watson value that was between 1 and 3
  3. values of the residuals are normally distributed. We looked at P-P plots where dots are closely aligned with this with this the line
  4. homoscedasticity assumption was also checked
  5. no influential cases biasing the. All Cook's values were below one (max=0.059 for QoL variable

We added the following to the statistical analysis part:

To ensure there were no multicollinearity, Pearson correlation coefficients were calculated to examine the relationships between the predictors. The coefficient ( min r =   and max r = ) suggested that the assumptions of multicollinearity were not violated.  Moreover, tolerance variance inflation factor (VIF) values did not indicate a violation of this assumption.

A Durbin-Watson statistic was calculated to assess the assumption that the values of residuals are independent, which suggested that this assumption was not violated in all models.

A Scatter plot was created to assess the assumption that the variance of the residuals was constant (homoscedasticity). Furthermore a P-P plot was created to assess the assumption that the values of the residuals are normally distributed. Also Cook's distance values were calculated to ensure that no influential cases were biasing the models.

  1. Lines 417-422 and 424-433: It is advisable to avoid duplication.

Answer:

Thank you for noting this, we changed accordingly.

  1. Appendix: Numeric values should appear in line X=1. The format of Table 5 can also be applied.

Answer:

Thank you for noting this, we changed accordingly.

  1. Table 1: Remove N and % in line "Menopausal Status". Information about other variables should also be presented in this table.

Answer:

Thank you, we removed N and % in line "Menopausal Status".

For the other variables, the information is just below Table 1 and we were advised not to duplicate information by the other reviewer.

  • Thank you for considering my opinion. I encourage authors to keep on working to improve the manuscript.

Answer:

We really appreciate the feedback, we believe after this round of reviews, our manuscript has improved a lot.

Round 2

Reviewer 1 Report

I have no further comments

Author Response

Thank you for taking the time to review our paper.